# Hepatitis A Seroprevalence Among HIV-Exposed and Unexposed Pediatric Populations in South Africa

**DOI:** 10.3390/vaccines12111276

**Published:** 2024-11-13

**Authors:** Edina Amponsah-Dacosta, Lufuno Ratshisusu, Lorato M. Modise, Ntombifuthi Blose, Omphile E. Simani, Selokela G. Selabe, Benjamin M. Kagina, Rudzani Muloiwa

**Affiliations:** 1Vaccines for Africa Initiative, School of Public Health, Faculty of Health Sciences, University of Cape Town, Cape Town 7925, South Africa; benjamin.kagina@uct.ac.za (B.M.K.); rudzani.muloiwa@uct.ac.za (R.M.); 2HIV and Hepatitis Research Unit, Department of Virology, Sefako Makgatho Health Sciences University, Pretoria 0208, South Africa; lratshisusu@gmail.com (L.R.); lorato.modise@smu.ac.za (L.M.M.); omphile.simani@smu.ac.za (O.E.S.); gloria.selabe@smu.ac.za (S.G.S.); 3Health Systems Trust, Durban 4001, South Africa; ntombifuthi.blose@hst.org.za; 4National Health Laboratory Service, Sefako Makgatho Health Sciences University, Pretoria 0208, South Africa; 5Department of Paediatrics and Child Health, Faculty of Health Sciences, University of Cape Town, Cape Town 7925, South Africa

**Keywords:** hepatitis A, hepatitis A virus, HIV, infants, maternal antibody, South Africa, water sanitation and hygiene

## Abstract

**Background:** There is limited evidence comparing hepatitis A seroprevalence among HIV-exposed uninfected (HEU), HIV-infected (HIV), and unexposed uninfected (HUU) children. This compromises rational vaccine decision-making. **Methods:** This study comprised a retrospective health facility-based population of children aged 1 month–12 years. Archival sera were tested for markers of acute (anti-HAV IgM) or past (total anti-HAV) HAV infection. Subgroup analysis was conducted based on perinatal HIV exposure or infection status. **Results:** Among 513 children, the median age was 10 (IQR: 4–25) months. The median maternal age was 29 (IQR: 25–34) years. An anti-HAV seropositivity of 95.1% (117/122 [95% CI 90.2–98.4]) was found among those ≤6 months of age, indicative of the rate of transplacental antibody transfer. Among 1–12-year-olds, hepatitis A seroprevalence was 19.3% (37/192 [95% CI 14.1–25.7]), while 1.1% (2/188 [95% CI 0.12–2.76]) had evidence of acute infection. Compared to HIV-exposed subgroups (HIV = 60%, 6/10 [95% CI 27.4–86.3] and HEU = 45%, 9/20 [95% CI 23.8–68]), hepatitis A seroprevalence among HUU children was low (29.2%, 47/161 [95% CI 22.4–37.0]). **Conclusions:** Natural immunity among HIV-exposed and unexposed children in South Africa is insufficient to protect against severe liver complications associated with HAV infection later in adulthood.

## 1. Introduction

The hepatitis A virus (HAV) is an important cause of acute viral hepatitis globally, accounting for 159 million cases of acute hepatitis A, 39,300 deaths, and 2.35 million disability-adjusted life years in 2019 alone [1,2,3]. The virus can be transmitted via the fecal–oral route through ingestion of contaminated food or water, or through close physical contact with an infectious person. Following infection with HAV, an incubation period ranging from 3 to 5 weeks is observed, characterized by high viremia and fecal viral shedding [4]. This is followed by the symptomatic phase, which typically presents as a self-limiting illness characterized by fever, abdominal pain, nausea, vomiting, or diarrhea. Acute hepatitis A is primarily diagnosed through the detection of anti-HAV immunoglobulin M (anti-HAV IgM) in serum using commercially available serologic assays [4,5]. It is important to note that on rare occasions (8–20%), anti-HAV IgM can be transiently detected among those with a history of hepatitis A vaccination [4]. While anti-HAV IgM can remain detectable for up to 6 months following initial infection, a neutralizing immunoglobulin G (IgG) response subsequently becomes dominant. Detection of anti-HAV IgG in the absence of IgM signals the resolution of the infection, with IgG providing lifelong immunity [4,5]. Pregnant women who have had previous exposure to HAV can provide passive immunity to their neonates via transplacental transfer of anti-HAV IgG. This ensures protection against early infection within the first 6 months of life, as maternally transferred IgG titers are known to wane from 7 months of age [6,7,8,9].

In rare (0.015–0.5%) severe cases, acute hepatitis A can progress into inflammatory liver disease and fulminant liver failure, which can be fatal [1,4,10]. Clinical outcomes are strongly correlated with age at infection; while infection in pediatric populations, particularly children under 6 years of age, are often (>90%) asymptomatic, the severity of disease and risk of fatality is increased (>70%) in older age groups [1,4,10]. Improved hygiene and sanitation, food and water safety, and vaccination are the mainstays of hepatitis A prevention and control [1].

Low- and middle-income countries with poor sanitary conditions and poor access to safe drinking water are disproportionately impacted by HAV infection, experiencing 66% and 97% of the global total of cases and deaths, respectively [1]. In endemic regions like parts of South Asia and sub-Saharan Africa, most children would have been exposed to the virus and developed lifelong immunity after recovery, characterized by ≥90% seroprevalence of anti-HAV IgG among the population by 10 years of age [10,11]. Emerging evidence suggests that South Africa is in transition from high to intermediate endemicity, with anti-HAV seroprevalence reported to only reach >90% among adults ≥ 25 years of age [11,12,13]. This is concerning, given that populations with intermediate endemicity experience higher incidence of severe clinical outcomes owing to the greater proportion of susceptible individuals who are at increased risk of symptomatic disease later in adulthood. While the World Health Organization recommends routine vaccination for children ≥ 12 months of age in countries with intermediate endemicity, the hepatitis A vaccine is currently not part of the South African Expanded Programme on Immunization schedule [1,14]. The decision to introduce the vaccine will have to be informed by cost considerations, the prioritization of other non-pharmaceutical interventions (improving sanitary conditions, and food and water safety) versus vaccine introduction, and the risk and burden of disease, among other contextual factors. For this purpose, National Immunization Technical Advisory Groups will require robust research evidence, including local epidemiological data among key risk groups, in order to make expert vaccine policy recommendations.

The disease epidemiology in South Africa is characterized by localized and widespread community and institutional outbreaks, with variations across provinces and socio-demographic and economic groups [14,15]. Passive laboratory-based surveillance data collected between 2017 and 2020 show a national hepatitis A incidence rate (based on IgM anti-HAV positivity) of 6–10 cases/100,000 per year among 1–9-year-olds, with the highest incidence rates recorded in the Western Cape province (7–10/100,000) [15]. In a previous nationwide seroprevalence study, it was reported that children 1–4 years of age comprised the highest proportion (33.5%) of acute HAV infections, with the highest positivity rates also recorded in the Western Cape [12]. Further cross-sectional and hospital-based studies have documented the seroprevalence (44.1% anti-HAV positivity in 1–7-year-olds), occurrence, and clinical outcomes (including acute liver failure and death) of symptomatic hepatitis A among children in the Western Cape, often compounded by underlying HIV infection [13,16,17]. Previous studies indicate that persons living with HIV tend to experience prolonged HAV viremia and extended periods of viral shedding, which could increase transmission risks. Furthermore, co-infection with HIV can lead to more severe disease outcomes [10,18]. Evidence on the seroprevalence of hepatitis A among children living with HIV is scarce. Only one recent hospital-based study assessed the seroprevalence of hepatitis A among children and adolescents (1–15 years) living with or without HIV in the Gauteng province, and found that while pediatric HIV infection and maternal HIV status were associated with a higher likelihood of testing positive for anti-HAV, these were not significant predictors of hepatitis A seropositivity [19].

There is a need for further epidemiological data to improve our limited understanding of the seroprevalence of hepatitis A in the context of pediatric HIV infection. Owing to the success of one of the world’s largest HIV programs aimed at the prevention of mother-to-child transmission (PMTCT), there is a growing population of HIV-exposed uninfected (HEU) children in South Africa. Of the 14.8 million HEU children in the world, 90% are from sub-Saharan Africa and 23.8% live in South Africa alone [20]. Due to potential immune and metabolic alterations following prolonged exposure to maternal HIV infection and antiretroviral prophylaxis in the intrauterine environment, HEU children are reported to experience greater susceptibility to and severity of common infectious causes of childhood morbidity and mortality compared to their HIV-unexposed uninfected (HUU) counterparts [21,22]. While the effects of early exposure to antiretroviral prophylaxis, particularly during the neonatal period, on the natural progression of hepatitis A remain underexplored, there is emerging evidence that immune reconstitution from Highly Active Antiretroviral Therapy may influence hepatitis A severity [10]. There is also evidence to suggest that HEU children may be at increased risk of other viral hepatitis infections [23,24,25]. Despite this, no other studies have been conducted to better understand whether the seroprevalence of hepatitis A is comparable among HEU children and those living with or without HIV in South Africa. Such evidence gaps compromise rational vaccine decision-making and the prioritization of targeted public health interventions. The aim of this study, therefore, is to determine the seroprevalence and risk factors of hepatitis A among children living with HIV and HEU children, compared to their HUU counterparts.

## 2. Materials and Methods

### 2.1. Study Design and Population

We conducted a retrospective cross-sectional study to determine the seroprevalence of hepatitis A among a conveniently sampled population of sera obtained from children receiving in- or out-patient care between 2012 and 2016 from health facilities within the Western Cape province of South Africa. These participants were part of an overarching study which investigated the incidence of pertussis among children presenting with lower respiratory tract infections [26]. Based on an expected hepatitis A seroprevalence of 44.1% among children in the Western Cape, a sample size of 162 provided a precision of 5% around the point estimate [13,27,28]. A total of 513 archival sera were available for this study. In addition, participants’ demographic (age, sex, ethnicity), clinical (maternal and child HIV status, vaccination history), and socio-economic (maternal level of education, type of dwelling, access to safe drinking water and other basic amenities) records were assessed. Based on perinatal HIV exposure and infection status, participants were stratified into HIV, HEU, and HUU subgroups. A fourth group of HIV-exposed participants whose infection status was unknown (HIV-E) was also assessed (Figure 1). Given the under-representation of HEU pediatric populations in research studies with this specific focus, this purposively sampled population presents a unique opportunity to better understand the burden of hepatitis A and the need for tailored interventions in a relatively under-served but growing population in our setting.

### 2.2. Serological and Molecular Testing for Hepatitis A

To detect the presence of acute and past HAV infections, serum samples were tested in 2023 for anti-HAV IgM and total anti-HAV (IgM/IgG), respectively, using Elecsys^®^ test kits (Roche Diagnostics, Mannheim, Germany). Serological assays were performed on the Cobas^®^ 6000 Analyzer (Roche Diagnostics International Ltd., Rotkreuz, Switzerland), which is a fully automated system used for heterogeneous immunoassays based on the electrochemiluminescence principle for qualitative and quantitative in vitro determinations. One sample could not be tested for either marker as the Cobas^®^ 6000 Analyzer detected an integrity issue (Figure 1). In line with the manufacturer’s instructions, samples with a cut-off index (COI) ≥ 1.0 were considered reactive for anti-HAV IgM, while the COI for a positive anti-HAV test was ≤1.0. As an exploratory objective, samples testing positive for IgM anti-HAV were further subjected to a reverse transcription–polymerase chain reaction (RT-PCR) to amplify the viral VP1-P2B gene fragment (393 bp) using a set of external (2870P and 3381N) and internal (2896P and 3289N) primers, as previously published [29,30]. Amplified products were confirmed against positive (supplied by the National Institute for Communicable Diseases of South Africa) and negative controls (PCR-grade H_2_O, Roche Diagnostics, Mannheim, Germany) by gel electrophoresis run on a 1% ethidium bromide-stained agarose gel.

### 2.3. Data Analysis

All statistical analyses were conducted using R version 4.3.1 and R studio version 2023.09.1+494 [31]. Data visualizations were conducted using Tableau software 2024.1.0 [32]. Maternal and child ages are described using medians with interquartile ranges (IQR). Proportions are presented as percentages with 95% confidence intervals (CI). Subgroup comparisons or associations between dichotomous variables were determined using Chi-square or Fisher’s exact tests depending on the expected cell count. The Kruskal–Wallis test was used to determine the differences between the medians of three or more independent groups as appropriate. Probable independent risk factors for past or existing HAV infections were determined by univariable logistic regression analysis. Associations are expressed as odds ratios (OR) with 95% CI. For missing data, we assumed that this occurred completely at random, as missingness did not depend on anything related to the research question and was thus unlikely to bias our analysis. We therefore conducted a complete record analysis [33].

## 3. Results

### 3.1. Characteristics of the Study Population

The demographic characteristics of the study population are presented in Table 1. Among the 513 participants in this study, there was an overrepresentation of males (54.0%, *n* = 277), children under the age of 1 year (53.8%, *n* = 276), and children of Black African ethnicity (65.7%, *n* = 337). Where crèche or pre-school attendance was concerned, 25.1% (*n* = 129) of the study population, ranging in age from 2 months to 12 years, had ever attended or were attending crèche at the time of participating in the overarching study. Most children (76.0%, *n* = 390) had been exclusively breastfed within the first 4 months of life. All mothers were screened for HIV infection in line with national antenatal guidelines, and based on maternal clinical records, 19.7% (101/513) had been diagnosed with HIV, 50.5% (51/101) of whom reported being on antiretroviral therapy. When stratified by perinatal HIV exposure, most (80.3%, *n* = 412) participants were HIV-unexposed and, as such, were categorized into the HUU subgroup (Table 1). The distribution of demographic characteristics among the HIV-exposed strata (HIV, *n* = 19; HEU, *n* = 74; and HIV-E, *n*= 8) is shown in Appendix A.

The median (IQR) maternal age was 29 (25–34) years, with most mothers (87.3% [448/513]) being ≥22 years at the time of enrolment in the overarching study (Table 2). The overwhelming majority of mothers (92.6% [475/513]) had received a secondary school education (grade 9–12) or higher (tertiary education including university), and this was comparable across the subgroups (Table 2). Based on available self-reported asset ownership (including type of dwelling, access to basic amenities), employment, and education, the study population (*N* = 513) could be categorized into low (20.3%, *n* = 104), lower-middle (6.6%, *n* = 34), upper-middle (48.1%, *n* = 247), and high (12.5%, *n* = 64) socio-economic quartiles. The HIV-exposed subgroup had a relatively higher proportion (25.7%, *n* = 26) of mothers in the low socio-economic quartile. In terms of access to basic amenities and sanitation, 64.1% (264/412) of the HUU subgroup lived in brick housing, while 66.4% (67/101) of their HIV-exposed counterparts lived in informal dwellings made of wood or shacks, tin, zinc, or iron sheeting (Appendix A). The highest proportion of those with access to indoor taps was found in the HUU subgroup (51.9% [214/412]), while the majority of the overall study population (84.8% [435/513]) had access to flush toilets, with limited use of pit latrines (13.1% [67/513]) or the bucket system (1.6% [8/513]) (Table 2).

### 3.2. Prevalence of Hepatitis A

Of the 512 eligible archival sera, 71.5% (*n* = 366) and 72.9% (*n* = 373) had sufficient volumes and could be tested for anti-HAV and anti-HAV IgM, respectively (Figure 1). Overall, 63.7% (326/512) could be tested for both anti-HAV and anti-HAV IgM, while 12.6% (47/373) of those tested for anti-HAV IgM could not be tested for anti-HAV, and 10.9% (40/366) of those tested for anti-HAV could not be tested for anti-HAV IgM. Of those tested, 56.6% (207/366 [95% CI 51.4–61.8]) were positive for anti-HAV while 0.5% (2/373 [95% CI 0.09–2.13]) had evidence of an acute HAV infection. Peak anti-HAV seropositivity (95.9%, 117/122 [95% CI 90.2–98.4]) was observed among those ≤6 months of age, suggestive of the rate of transplacental transfer of anti-HAV. Excluding <1-year-olds with possible persistence of passively transferred anti-HAV, seropositivity for anti-HAV and anti-HAV IgM in those 1–12 years of age was 19.3% (37/192 [95% CI 14.1–25.7]) and 1.1% (2/188 [95% CI 1.29–3.79]), respectively. The seroprevalence of hepatitis A declined with increasing age, reaching 40.0% (2/5 [95% CI 5.27–85.3]) by age 10 years and older (Figure 2).

The two cases of acute HAV infection were detected in a 2- and 3-year-old who were in the HIV and HEU subgroups, respectively. Of these two cases, only one participant had sufficient serum for detection of viral RNA using RT-PCR. This participant was in the convalescent phase of infection (tested positive for both anti-HAV IgM and anti-HAV) characterized by low or undetectable viremia. Consequently, the RT-PCR was negative and the viral VP1-P2B gene fragment could not be amplified.

When comparing those 1–12 years of age with (HIV, HEU, and HIV-E subgroups) and without (HUU subgroup) perinatal HIV exposure, the proportion of those with natural immunity was highest among the HIV-exposed population. Seropositivity for anti-HAV was 51.6% (16/30) and 29.2% (47/161) in the HIV-exposed subgroup compared to the unexposed subgroup, respectively (*p* = 0.015) (Figure 3). That for anti-HAV IgM was 6.7% (2/30) in the HIV-exposed subgroup, with no cases of acute infection detected among HUU children. The highest proportion (70.8%, 114/161) of susceptible children was found in the HUU subgroup (*p* = 0.015) (Figure 3). The seropositivity values for anti-HAV across subgroups and age strata are presented in Figure 3 and Figure 4.

Anti-HAV seropositivity followed similar trends regardless of perinatal HIV exposure or infection status, declining after the first 6 months of life and peaking among those ≥5 years, but never reaching higher than 50% among those 10 years and older (Figure 4). Among HIV-exposed children, anti-HAV seropositivity was comparably higher in the HIV subgroup (60%, 6/10 [95% CI 27.4–86.3]) than in the HEU subgroup (45%, 9/20 [95% CI 23.8–68]).

### 3.3. Risk Factors of Hepatitis A Seropositivity

We assessed potential independent risk factors of hepatitis A seropositivity among study participants who tested positive for at least one marker (anti-HAV and/or anti-HAV IgM). Overall, *n* = 206 participants tested positive for anti-HAV alone, *n* = 1 for both anti-HAV and anti-HAV IgM, and *n* = 1 for anti-HAV IgM only. Thus, *n* = 208 participants were included in the logistic regression analysis (Appendix A). Compared to <1 year-olds, children between the ages of 1 and 4 years had a lower likelihood of being seropositive, with an OR of 0.23 [95% CI 0.14–0.36]. Further regression analysis did not identify additional significant associations with other socio-demographic or economic variables, including maternal HIV exposure.

## 4. Discussion

This study reports evidence on the seroprevalence of hepatitis A among a pediatric population of HIV-exposed infected and uninfected children, compared to their HIV-unexposed uninfected counterparts, in South Africa. Overall, age-specific trends in anti-HAV seropositivity appear to be comparable among HIV-exposed and HIV-unexposed children. Perinatal HIV exposure was not found to be an independent risk factor for hepatitis A seropositivity in early childhood. Our findings are consistent with others suggesting an intermediate hepatitis A endemicity (<90% anti-HAV positivity by age 10 years) in South Africa [6,7,8]. The implication of this is that the level of natural immunity within the population remains insufficient to protect against symptomatic HAV infection and severe clinical outcomes later in adulthood without timely and appropriate public health intervention.

We found that among ≤6-month-olds, anti-HAV seropositivity was near-universal at 95.1% and could serve as a proxy for the seroprevalence of hepatitis A among pregnant women and mothers in our study setting. The detection of anti-HAV among newborns is associated with transplacental transfer of antibodies which provide temporary protection against HAV infection [6]. Concentrations of passively transferred maternal antibodies remain high during the first 6 months of life but wane significantly between months 7 and 12 [6,7,8,9]. This was also observed in our study, demonstrated by a decline in anti-HAV seropositivity among 7–11-month-old participants across all subgroups. Cases of acute HAV infection were only observed after the first year of life—which coincides with the decline in protective maternal anti-HAV—and only among children living with perinatal HIV exposure. In a hospital-based hepatitis A seroprevalence study conducted between 2018 and 2019, du Plessis et al. report an anti-HAV IgM seroprevalence of 2.62% among 1–15-year-olds [19]. Although they do not disaggregate this seroprevalence rate between those living with and without HIV infection, it is lower than the 6.7% anti-HAV IgM seroprevalence identified in our HIV-exposed subgroup of 1–12-year-olds. With cases detected in 2- and 3-year-olds, our findings are also consistent with the peak age range (1–4 years) for acute HAV infection among children in South Africa [12]. The public health significance of HAV–HIV co-infection lies in its association with a longer duration of HAV viremia and shedding in stool, increasing the likelihood of onward spread in settings with poor sanitation and hygiene practices [10,18].

In a hospital-based study conducted in São Paulo, Brazil, assessing the prevalence of hepatitis A antibodies in children born to mothers living with HIV, Gouvêa et al. found an overall hepatitis A seroprevalence of 26%, with a higher seroprevalence in children living with HIV (35.5%) than in HEU children (16.7%) aged 2–10 years [34]. However, this study did not include an HUU subgroup. Among the 1–12-year-olds in our study, the seroprevalence of hepatitis A was 19.3%, with a relatively higher prevalence rate in the HIV subgroup (60%) than in the HEU (45%) subgroup, although the HUU subgroup (29.2%) had the lowest proportion of participants testing positive for anti-HAV overall. The hepatitis A seroprevalence rate in our study (19.3%) is considerably lower than the 44.1% reported among 1–7-year-olds (HIV status unknown) in 2015 in the Western Cape province [13]. A recent systematic review by Patterson et al. estimates that the seroprevalence of hepatitis A among children and adolescents (1–18 years) in Africa is 57%, consistent with an intermediate endemicity [11]. While no other studies have assessed the seroprevalence of hepatitis A among HEU children in South Africa, one other study conducted in children (1–15 years) living with and without HIV found rates ranging between 24.0 and 85.7% and between 18.6 and 100%, respectively [19]. Differences in hepatitis A seroprevalence rates may be associated with variable socio-economic status among study populations. In the du Plessis et al. study, increased age and living in informal dwellings, rather than HIV status, were statistically associated with HAV seropositivity [19]. In our study, the level of maternal education was high, and most participants were in the upper-middle socio-economic quartile, lived in formal housing, and had good access to safe drinking water (indoor taps) and sewage disposal systems (flush toilets). The socio-economic characteristics of our study population are consistent with findings from the national household survey [35,36]. Previous studies have shown that common risk factors for HAV infection in other low- and middle-income countries include a low level of knowledge about hepatitis, low family income, over-crowding, poor hygiene practices or sanitary conditions, and poor access to safe drinking water [10,37,38].

The findings of this study should be considered in light of some limitations, including the retrospective study design, the modest sample size across HIV and age subgroups, and the use of a hospital-based population who may have risk profiles and health-seeking behaviors that are not representative of the general population. Despite this, we contribute crucial evidence to the limited body of knowledge on the seroprevalence and risk factors of hepatitis A among HIV-exposed infected and uninfected children. Future research directions should include prospective studies in community-representative populations in order to enhance the generalizability of findings and guide public health policy and practice.

## 5. Conclusions

The findings of this study underscore the intermediate endemicity of hepatitis A in South Africa even in the context of perinatal HIV exposure. We show a high level of maternal transfer of anti-HAV among children under 6 months of age regardless of perinatal HIV exposure, with acute HAV infections only observed among HIV-exposed 1–4-year-olds. Overall, the seroprevalence of hepatitis A among children between 1 and 12 years of age was 19.3%, and trends in anti-HAV seropositivity were comparable in HIV-exposed infected and uninfected, as well as unexposed uninfected children. There is a need to scale-up public health interventions given the high susceptibility rate within the population and the risk of localized and widespread outbreaks in South Africa. Such interventions should include intensive risk communication to the public, including pregnant women, and parents and caregivers. In this regard, PMTCT services could also serve as useful entry points for raising awareness about hepatitis A and improving adherence to good hygiene practices and prevention and control strategies. Relevant government institutions should increase the provision of safe water, optimize wastewater treatment and management, and improve the socio-economic conditions within all communities as a matter of national priority in order to reduce the risk of hepatitis A outbreaks. Further to this, when making evidence-based decisions on whether to introduce the hepatitis A vaccine, African-based National Immunization Technical Advisory Groups should include epidemiological data on the seroprevalence of hepatitis A among the growing population of HEU children.

## Figures and Tables

**Figure 1 vaccines-12-01276-f001:**
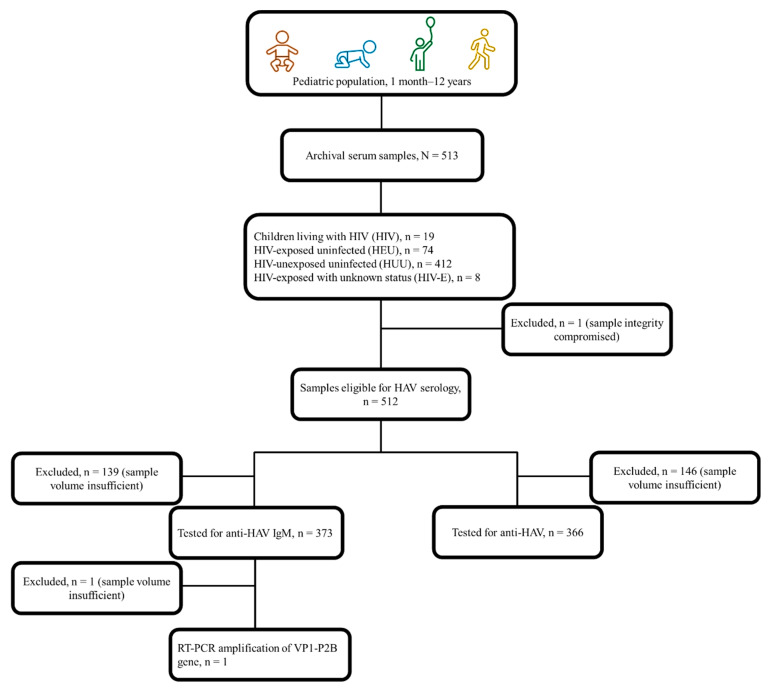
Sample stratification and laboratory testing for serological markers and viral RNA.

**Figure 2 vaccines-12-01276-f002:**
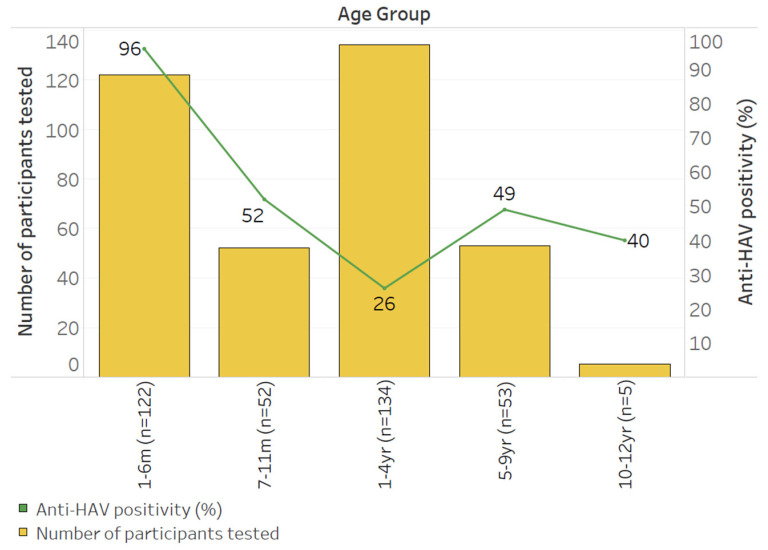
Seropositivity of anti-HAV stratified by age across the study population.

**Figure 3 vaccines-12-01276-f003:**
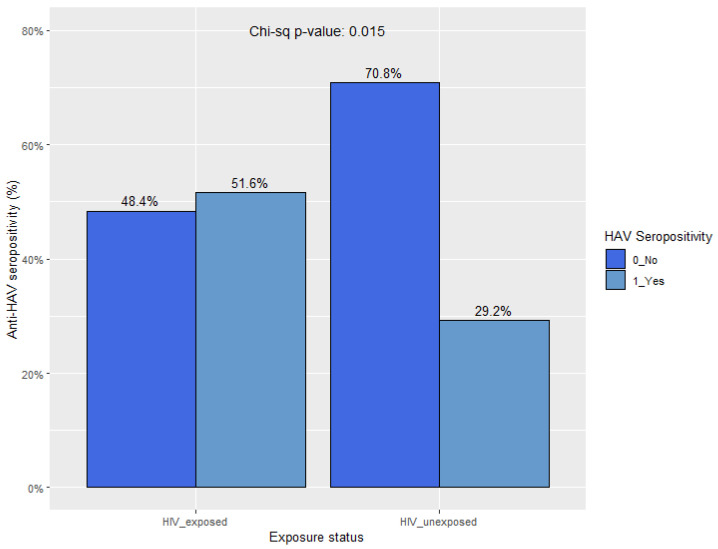
Anti-HAV seropositivity in HIV-exposed (*n* = 30) vs. -unexposed (*n* = 161) children 1–12 years of age.

**Figure 4 vaccines-12-01276-f004:**
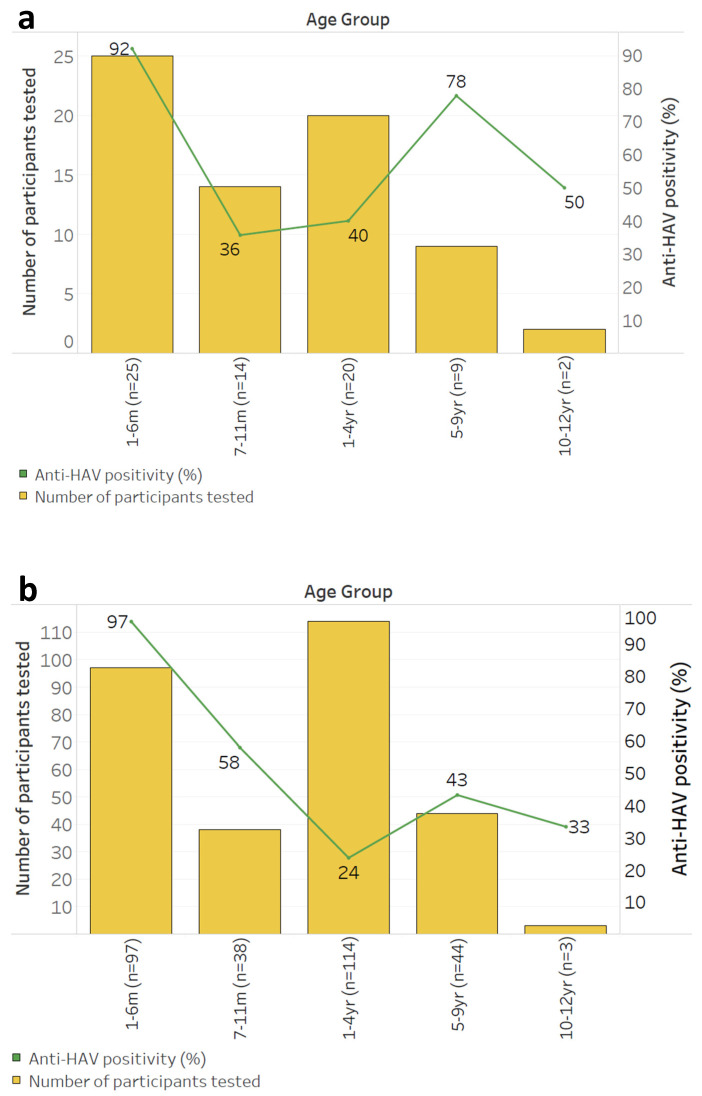
Seropositivity of anti-HAV stratified by age among (**a**) HIV-exposed vs. (**b**) HIV-unexposed children.

**Table 1 vaccines-12-01276-t001:** Participants’ socio-demographic characteristics stratified by perinatal HIV exposure.

Characteristics	Total, *N* = 513 (%)	HIV-Exposed ^1^, *n* = 101	HIV-Unexposed ^2^, *n* = 412
**Sex**			
*Male*	277 (54.0%)	59 (58.4%)	218 (52.9%)
*Female*	236 (46.0%)	42 (41.6%)	194 (47.1%)
**Age**			
*Median Age (IQR) months*	10 (4, 25)	10 (4, 19)	10 (4, 26)
*<1 year*	276 (53.8%)	56 (55.4%)	220 (53.4%)
*1–4 years*	177 (34.5%)	33 (32.7%)	144 (35.0%)
*5–12 years*	60 (11.7%)	12 (11.9%)	48 (11.7%)
**Ethnicity**			
*Black*	337 (65.7%)	96 (95.0%)	241 (58.5%)
*Colored/Mixed ancestry*	164 (32.0%)	3 (3.0%)	161 (39.1%)
*Other ethnicity*	1 (0.2%)	0 (0%)	1 (0.2%)
*Unknown* ^3^	11 (2.1%)	2 (2.0%)	9 (2.2%)
**Crèche attendance ^4^**			
*Yes*	129 (25.1%)	28 (27.7%)	101 (24.5%)
*No*	380 (74.1%)	72 (71.3%)	308 (74.8%)
*Unknown* ^3^	4 (0.8%)	1 (1.0%)	3 (0.7%)
**Feeding at ≤4 months**			
*Breast milk*	390 (76.0%)	68 (67.3%)	322 (78.2%)
*Breast milk and formula*	108 (21.1%)	32 (31.7%)	76 (18.4%)
*Formula (never breastfed)*	14 (2.7%)	1 (1.0%)	13 (3.2%)
*Unknown* ^3^	1 (0.2%)	0 (0%)	1 (0.2%)
**Feeding at >4 months**			
*Breast milk*	9 (1.8%)	1 (1.0%)	8 (1.9%)
*Breast milk and formula*	468 (91.2%)	88 (87.1%)	380 (92.2%)
*Formula (never breastfed)*	11 (2.1%)	2 (2.0%)	380 (92.2%)
*Unknown* ^3^	25 (4.9%)	10 (10.0%)	15 (3.6%)

^1^ HIV-exposed includes children living with HIV, HIV-exposed uninfected, and HIV-exposed but child infection status unknown; ^2^ HIV-unexposed uninfected children; ^3^ missing data; ^4^ preschool.

**Table 2 vaccines-12-01276-t002:** Maternal socio-demographic and economic characteristics stratified by perinatal HIV exposure.

Characteristics	Total, *N* = 513 (%)	HIV-Exposed, *n* = 101	HIV-Unexposed, *n* = 412
**Age (years)**			
*Median Age (IQR)*	29 (25.34)	31 (26.34)	28 (24.34)
*Teen (15–19)*	33 (6.4%)	4 (4.0%)	29 (7.0%)
*Young (20–21)*	32 (6.2%)	4 (4.0%)	28 (6.8%)
*Adult (22–51)*	448 (87.3%)	93 (92.0%)	355 (86.2%)
**Education level**			
*Primary*	20 (3.9%)	5 (5.0%)	15 (3.6%)
*Secondary*	460 (89.7%)	91 (90.0%)	369 (90.0%)
*Highest basic school grade unknown* ^1^	18 (3.5%)	4 (4.0%)	14 (3.4%)
*Tertiary*	15 (2.9%)	1 (1.0%)	14 (3.4%)
**Type of dwelling**			
*Bricks*	298 (58.1%)	34 (33.7%)	264 (64.1%)
*Shack (tin/zinc/iron sheeting)*	196 (38.2%)	64 (63.4%)	132 (32.0%)
*Wood or Wendy house*	16 (3.1%)	3 (3.0%)	13 (3.2%)
*Unknown* ^2^	3 (0.6%)	0 (0%)	3 (0.7%)
**Water source**			
*Indoor tap*	239 (46.6%)	25 (24.8%)	214 (51.9%)
*Outdoor communal tap*	110 (21.4%)	30 (29.7%)	80 (19.4%)
*Borehole*	164 (32%)	46 (45.5%)	118 (28.6%)
**Type of toilet**			
*Flush toilet*	435 (84.8%)	76 (75.2%)	359 (87.1%)
*Bucket*	8 (1.6%)	3 (3.0%)	5 (1.2%)
*Pit latrine*	67 (13.1%)	21 (20.8%)	46 (11.2%)
*Unknown* ^2^	3 (0.6%)	1 (1.0%)	2 (0.5%)
**Socio-economic quartiles ^3^**			
*Low*	104 (20.3%)	26 (25.7%)	78 (18.9%)
*Lower-middle*	34 (6.6%)	2 (2.0%)	32 (7.8%)
*Upper-middle*	247 (48.1%)	50 (49.5%)	197 (47.8%)
*High*	64 (12.5%)	12 (11.9%)	52 (12.6%)
*Unknown* ^2^	64 (12.5%)	11 (10.9%)	53 (12.9%)

^1^ Participants indicated having received a basic education (primary or secondary), but highest grade attained was not recorded; ^2^ missing data; ^3^ socio-economic status was categorized into quartiles based on a validated weighted composite score that included maternal asset ownership, employment, and education.

## Data Availability

The datasets used and analyzed during the current study are available from the corresponding author on reasonable request.

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
