# Peer review of "Hepatitis A Seroprevalence Among HIV-Exposed and Unexposed Pediatric Populations in South Africa"

_vaccines, 2024, doi:10.3390/vaccines12111276_

Round 1

Reviewer 1 Report

Comments and Suggestions for Authors

Hepatitis A seroprevalence among HIV-exposed and unexposed pediatric populations in South Africa is limited. This study includes data from 513 pediatric participants with and without HIV exposure and demonstrates that hepatitis A seroprevalence is higher among HIV-exposed infected children (60%) and HIV-exposed uninfected children (45%) compared to HIV-unexposed uninfected children (29%) aged 1–12 years, after maternal transferred IgG has been cleared. This study has clinical significance and underscores the need for routine HAV vaccination in developing countries.

Minor comments

1.     Introduction: Please clarify why anti-HAV IgM represents acute infection while total anti-HAV (IgM/IgG) represents past HAV infection and provide related references.

2.       Figure 1: please use unisex carton to replace the woman carton unless the older 5-12 years old children were all female.

3.       Lines 132 to 134: Please clarify the COI for a positive anti-HAV. It could not be ≤1.0. For example, COI=0 or COI<0 could not be considered reactive although both zero and negative numbers are <1.

4.       Inconsistent information between lines168-169 and Table 1. In lines 168-169, the authors said that “Most children (76.8%, n=394) had received mixed feeding (breast and formula fed) within the first 4 months of life”, but in Table 1, the authors showed that the Feeding at ≤4 months by breast milk and formula is 108 (21.1%). Please make the results consistent.

5.       Figure 2 and 4: please use period instead of comma for the numbers.

Author Response

We thank the Reviewer for lending their time and expertise to this peer-review process. The feedback provided has been crucial in improving the quality of our work as presented in the revised manuscript. We provide detailed responses to each of the comments raised. All line numbers referred to herein can be located in the “mark-up” version with revisions in track changes. We trust that you find all in order.

Reviewer’s Comment 1: Introduction: Please clarify why anti-HAV IgM represents acute infection while total anti-HAV (IgM/IgG) represents past HAV infection and provide related references.

Authors’ Response 1: We thank the Reviewer for pointing this out. We have now updated the Introduction section to include a statement clarifying the diagnostic significance of anti-HAV IgM and IgG in the natural history of hepatitis A. Appropriate references have also been provided. This revision can be found in lines 40 – 52 which read as follows:

Following infection with HAV, an incubation period ranging from 3 – 5 weeks is observed, characterized by high viremia and faecal viral shedding [4]. This is followed by the symptomatic phase which typically presents as a self-limiting illness characterized by fever, abdominal pain, nausea, vomiting, or diarrhoea. Acute hepatitis A is primarily diagnosed through the detection of anti-HAV immunoglobulin M (anti-HAV IgM) in serum using commercially available serologic assays [4,5]. It is important to note that on rare occasions (8 – 20%) anti-HAV IgM can be transiently detected among those with a history of hepatitis A vaccination. While anti-HAV IgM can remain detectable for up to 6 months following initial infection, a neutralizing immunoglobulin G (IgG) response subsequently becomes dominant. Detection of anti-HAV IgG in the absence of IgM signals the resolution of the infection, with IgG providing lifelong immunity [4,5].”  

Reviewer’s Comment 2: Figure 1: please use unisex carton to replace the woman carton unless the older 5-12 years old children were all female.

Authors’ Response 2: A unisex cartoon is now provided in Figure 1 as shown in lines 165 – 166. The cartoons do not represent any specific age or sex strata.

Reviewer’s Comment 3: Lines 132 to 134: Please clarify the COI for a positive anti-HAV. It could not be ≤1.0. For example, COI=0 or COI<0 could not be considered reactive although both zero and negative numbers are <1.

Authors’ Response 3: The Reviewer’s comment is well noted. We have crossed checked with the manufacturer’s factsheet (attached separately) which provide guidance for the interpretation of anti-HAV IgM and anti-HAV (IgM/IgG). The following guidance is provided by the manufacturer:

“Elecsys® Anti-HAV IgM assay interpretation: COI <1.0 = non-reactive | COI ≥1.0 = reactive”

“Elecsys® Anti-HAV II assay interpretation: COI >1.0 = non-reactive (negative for HAV-specific antibodies) | COI ≤1.0 = reactive (positive for HAV-specific antibodies)”

We have maintained this interpretation within the manuscript and trust that the Reviewer finds this in order.

Reviewer’s Comment 4: Inconsistent information between lines 168-169 and Table 1. In lines 168-169, the authors said that “Most children (76.8%, n=394) had received mixed feeding (breast and formula fed) within the first 4 months of life”, but in Table 1, the authors showed that the Feeding at ≤4 months by breast milk and formula is 108 (21.1%). Please make the results consistent.

Authors’ Response 4: We thank the Reviewer for bringing this error to our attention. This has been resolved to ensure that the text aligns with the data presented in Table 1. The revision has been made in lines 189 – 190 which now read as follows:

Most children (76.0%, n=390) had been exclusively breastfed within the first 4 months of life.”

Reviewer’s Comment 5: Figure 2 and 4: please use period instead of comma for the numbers.

Authors’ Response 5: We have recreated Figures 2 and 4 to remove the decimal points (see line 240 and 261, respectively). Unfortunately, the visualization tool (Tableau software 2024) used to generate the graphs for Figures 2 and 4 automatically presents decimal points using commas. For this reason, proportions have now been presented as whole numbers in Figures 2 and 4.

Reviewer 2 Report

Comments and Suggestions for Authors

            The authors evaluated the prevalence of antibodies against hepatitis A virus (HAV) in HIV-exposed and unexposed paedriatic population from South Afriva. They found that in all groups, the seroprevalence was too low, which warrants to include vaccination. Some concerns need to be attended before acceptance.

1.  The rationale of comparing the HAV seroprevalence between HIV exposed and unexposed children needs to be explained in more detail. The authors should describe if there is evidence of a more severe HAV disease in people living with HIV, and particularly if the HAART exposure in the neonatal period could affect the natural course of this disease?

2.  Very few samples are available for some groups, as people living with HIV (n=19).

3.  Table 1. Why HIV-exposed infected and non-infected were mixed in this table? The authors describe different cohort of patients and then mix them.

4.  Figure 3 brings few information.

Author Response

We thank the Reviewer for lending their time and expertise to this peer-review process. The feedback provided has been crucial in improving the quality of our work as presented in the revised manuscript. We provide detailed responses to each of the comments raised. All line numbers referred to herein can be located in the “mark-up” version with revisions in track changes. We trust that you find all in order.

Reviewer’s Comment 1: The rationale of comparing the HAV seroprevalence between HIV exposed and unexposed children needs to be explained in more detail. The authors should describe if there is evidence of a more severe HAV disease in people living with HIV, and particularly if the HAART exposure in the neonatal period could affect the natural course of this disease?

Authors’ Response 1: The Reviewer’s comment is well noted. We provide rationale for comparing HAV seroprevalence between HIV-exposed (HIV and HEU) and unexposed children (HUU), as well as the potential influence of HAART exposure on the natural course of HAV infection in lines 93 – 124 which now read as follows:

Further cross-sectional and hospital-based studies have documented the seroprevalence (44.1% anti-HAV positivity in 1 – 7-year-olds), and the occurrence and clinical outcomes (including acute liver failure and death) of symptomatic hepatitis A among children in the Western Cape, often compounded by underlying HIV infection [9,12,13]. Previous studies indicate that persons living with HIV tend to experience prolonged HAV viremia and extended periods of viral shedding, which could increase transmission risks. Furthermore, co-infection with HIV can lead to more severe disease outcomes [6,14]. Evidence on the seroprevalence of hepatitis A among children living with HIV is scarce. Only one recent hospital-based study assessed the seroprevalence of hepatitis A among children and adolescents (1 – 15 years) living with or without HIV in the Gauteng province and found that while paediatric HIV infection and maternal HIV status were associated with a higher likelihood of testing positive for anti-HAV, these were not significant predictors of hepatitis A seropositivity [15]. 

There is a need for further epidemiological data to improve our limited understanding of the seroprevalence of hepatitis A in the context of paediatric HIV infection. Owing to the success of one of the world’s largest HIV programs aimed at prevention of mother-to-child transmission (PMTCT), there is a growing population of HIV-exposed uninfected (HEU) children in South Africa. Of the 14.8 million HEU children in the world, 90% are from sub-Saharan Africa and 23.8% live in South Africa alone [16]. Due to potential immune and metabolic alterations following prolonged exposure to maternal HIV infection and antiretroviral prophylaxis in the intrauterine environment, HEU children are reported to experience greater susceptibility and severity of common infectious causes of childhood morbidity and mortality compared to their HIV-unexposed uninfected (HUU) counterparts [17,18]. While the effects of early exposure to antiretroviral prophylaxis, particularly from the neonatal period, on the natural progression of hepatitis A remain underexplored, there is emerging evidence that immune reconstitution from Highly Active Antiretroviral Therapy may influence hepatitis A severity [6]. There is also evidence to suggest that HEU children may be at increased risk of other viral hepatitis infections [19–21]. Despite this, no other studies have been conducted to better understand whether the seroprevalence of hepatitis A is comparable among HEU children and those living with or without HIV in South Africa. Such evidence gaps compromise rational vaccine decision-making and prioritization of targeted public health interventions.”  

Reviewer’s Comment 2: Very few samples are available for some groups, as people living with HIV (n=19).

Authors’ Response 2: Indeed, we identified very few perinatal HIV infections within our population. This can be attributed to the success of prevention of mother-to-children-transmission (PMTCT) programs in South Africa. We note the modest sample sizes across HIV and age strata as part of the study limitations section in lines 336 – 339, which read as follows:

The findings of this study should be considered in light of some limitations including the retrospective study design, the modest sample size across HIV and age subgroups, and the use of a hospital-based population who may have risk profiles and health seeking behaviours that are not representative of the general population.”

Reviewer’s Comment 3: Table 1. Why HIV-exposed infected and non-infected were mixed in this table? The authors describe different cohort of patients and then mix them.

Authors’ Response 3: The primary intent of this study was to determine the seroprevalence and risk factors of hepatitis A among HIV-exposed and unexposed children. We recognise that HIV-exposed children do include those living with HIV (HIV) and those who are uninfected (HEU), and we make this distinction accordingly. Per the Reviewer’s second comment (see Comment 2), there were limited numbers of participants in the HIV and HEU sub-groups compared to the HUU. For Table 1 we present demographic data for HIV-exposed vs unexposed to enhance comparability. A detailed breakdown of demographic data for HIV, HEU vs HUU children can be found in Supplementary Tables 1 and 2.

Reviewer’s Comment 4: Figure 3 brings few information.

Authors’ Response 4: We thank the Reviewer for this comment. Figure 3 presents a comparison of HAV seropositivity among 1 – 12-year-olds within the HIV-exposed vs unexposed sub-groups. The narrative accompanying Figure 3 has been updated for clarity in lines 242 – 249 and now read as follows:

When comparing those 1 – 12 years of age with (HIV, HEU, and HIV-E subgroups) and without (HUU subgroup) perinatal HIV exposure, the proportion of those with natural immunity was highest among the HIV-exposed population. Seropositivity of anti-HAV was 51.6% (16/30) and 29.2% (47/161) in the HIV-exposed compared to the unexposed subgroup, respectively (p = 0.015) (Figure 3). That for anti-HAV IgM was 6.7% (2/30) in the HIV-exposed subgroup with no cases of acute infection detected among HUU children. The highest proportion (70.8%, 114/161) of susceptible children was found in the HUU subgroup (p = 0.015) (Figure 3).”

Reviewer 3 Report

Comments and Suggestions for Authors

 The article presents interesting data on the immune status (anti-HIV) of HIV-exposed and unexposed children in South Africa. The study provides interesting data on the prevalence of hepatitis A and provides guidance for anti-HAV vaccination programs among different groups of children.

My main objections relate to the presentation of the data. You have actually tested for anti-HAV only 373 samples. You did not divide these 373 sera into groups (HIV, HEU, HUU), which greatly complicates understanding and changes the statistical analyses. 

 I think that even in the abstract you have to explain the term HIV exposure children or to change it to perinatal HIV exposure. You must also give the data by group (HIV, HEU, HUU) for the 513 sera you examined, and data of real analysed samplaes for anti-HAV. The first sentence in the abstract is difficult to understand and my advice is to rewrite it. 54% (277) were male, which is not important information. The abstract should be completely rewritten, giving more information about HIV exposure children and the data by groups of the samples you studied.

You have actually examined 373 samples due to technical limitations, it is necessary to make a distribution by groups of these 373 sera (HIV, HEU, HUU) again.

 L 172. I do not understand this sentence, please provide more information. When stratified by perinatal HIV exposure, most participants fell in the HUU (80.3%, n=412) subgroup.

In the introduction, it is essential to provide more information about the transmission of antibodies from mother to fetus in infancy and to give data from other studies. In the introduction I did not see data on similar studies in other countries, if they exist please provide this information.

The explanation of figure 3 is not clear. Please give more details. 

The quality of figure 4 is not good, and numbers are not visible well; please provide a higher resolution figure.

The discussion is interestingly written.

Author Response

We thank the Reviewer for lending their time and expertise to this peer-review process. The feedback provided has been crucial in improving the quality of our work as presented in the revised manuscript. We provide detailed responses to each of the comments raised. All line numbers referred to herein can be located in the “mark-up” version with revisions in track changes. We trust that you find all in order.

Reviewer’s Comment 1: My main objections relate to the presentation of the data. You have actually tested for anti-HAV only 373 samples. You did not divide these 373 sera into groups (HIV, HEU, HUU), which greatly complicates understanding and changes the statistical analyses.

Authors’ Response 1: The Reviewer’s comment is well noted. To provide clarity, we had a total of 512 eligible sera samples. Of this, 373 could be tested for anti-HAV IgM while 366 were tested for anti-HAV (total IgM/IgG). Not all samples could be tested for both. In fact, 326/512 could be tested for both anti-HAV IgM and anti-HAV. In addition to the clear breakdown of samples tested as shown in Figure 1, we provide detailed description of the samples tested in lines 221 – 225 which reads as follows:

Of the 512 eligible archival sera, 71.5% (n=366) and 72.9% (n=373) had sufficient volumes and could be tested for anti-HAV and anti-HAV IgM, respectively (Figure 1). Overall, 63.7% (326/512) could be tested for both anti-HAV and anti-HAV IgM, while 12.6% (47/373) of those tested for anti-HAV IgM could not be tested for anti-HAV and 10.9% (40/366) of those tested for anti-HAV could not be tested or anti-HAV IgM.

For this reason, it is not possible to start our analysis from a sample size of 373 sera as this will not be representative of those tested for anti-HAV and/or anti-HAV IgM. Thus, we provide a comprehensive breakdown of the seroprevalence of anti-HAV and anti-HAV IgM among the actual sera tested; first in the HIV-exposed (HIV and HEU) versus the unexposed (HUU), followed by a comparison across HIV, HEU, and HUU subgroups. We trust that this brings clarity, and the Reviewer finds the information provided in order.  

Reviewer’s Comment 2: I think that even in the abstract you have to explain the term HIV exposure children or to change it to perinatal HIV exposure. You must also give the data by group (HIV, HEU, HUU) for the 513 sera you examined, and data of real analysed samplaes for anti-HAV. The first sentence in the abstract is difficult to understand and my advice is to rewrite it. 54% (277) were male, which is not important information. The abstract should be completely rewritten, giving more information about HIV exposure children and the data by groups of the samples you studied.

Authors’ Response 2: We have now amended the abstract section to reflect the Reviewer’s suggestions. The first sentence has been revised for better clarity. Categorization of subgroups based on perinatal HIV exposure or infection has been emphasized, and the study sub-groups (HIV, HEU, and HUU) have been clearly identified. Distribution of demographic data (i.e., sex of participants) has been excluded. Based on our response to the Reviewer’s first comment (see Comment 1) regarding variability in samples tested for anti-HAV versus anti-HAV IgM, we have maintained the correct samples sizes tested per subgroup. The revised abstract can be found in lines 15 – 31, and read as follows:

There is limited evidence comparing hepatitis A seroprevalence among HIV-exposed uninfected (HEU), HIV infected (HIV), and unexposed uninfected (HUU) children. This compromises rational vaccine decision-making. This study comprised of a retrospective health facility-based population of children aged 1 month – 12 years. Archival sera were tested for markers of acute (anti-HAV IgM) or past (total anti-HAV) HAV infection. Subgroup analysis was conducted based on perinatal HIV exposure or infection status. Among 513 children, the median age was 10 (IQR: 4 – 25) months. The median maternal age was 29 (IQR: 25 – 34) years. An anti-HAV seropositivity of 95.1% (117/122 [95% CI 90.2-98.4]) was found among those ≤6 months of age, indicative of the rate of transplacental antibody transfer. Among 1–12-year-olds, hepatitis A seroprevalence was 19.3% (37/192 [95% CI 14.1-25.7]), while 1.1% (2/188 [95% CI 0.12-2.76]) had evidence of acute infection. Compared to HIV-exposed subgroups (HIV=60%, 6/10 [95% CI 27.4-86.3] and HEU=45%, 9/20 [95% CI 23.8-68]), hepatitis A seroprevalence among HUU children was low (29.2%, 47/161 [95% CI 22.4-37.0]). Natural immunity among HIV-exposed and unexposed children in South Africa is insufficient to protect against severe liver complications associated with HAV infection later in adulthood.”

Reviewer’s Comment 3: You have actually examined 373 samples due to technical limitations, it is necessary to make a distribution by groups of these 373 sera (HIV, HEU, HUU) again.

Authors’ Response 3: We thank the Reviewer for raising this issue once again, and trust that we have now provided clarity on this issue.

Reviewer’s Comment 4: L 172. I do not understand this sentence, please provide more information. When stratified by perinatal HIV exposure, most participants fell in the HUU (80.3%, n=412) subgroup.

Authors’ Response 4: This sentence has now been revised for clarity in lines 193 – 195 and reads as follows:

When stratified by perinatal HIV exposure, most (80.3%, n=412) participants were HIV-unexposed and as such, categorized into the HUU subgroup.”

Reviewer’s Comment 5: In the introduction, it is essential to provide more information about the transmission of antibodies from mother to fetus in infancy and to give data from other studies. In the introduction I did not see data on similar studies in other countries, if they exist please provide this information.

Authors’ Response 5: We thank the Reviewer for raising this and agree that such information in the Introduction section will better contextualize our findings. While we appraise published evidence on transplacental transfer of anti-HAV in our Discussion section (see lines 291 – 295), we have now included information (with appropriate references) on this as part of the revised Introduction section in lines 52 – 56 which read as follows:

Pregnant women who have had previous exposure to HAV can provide passive immunity to their neonates via transplacental transfer of anti-HAV IgG. This ensures protection against early infection within the first 6 months of life, as maternally transferred IgG titres are known to wane from 7 months of age [6–9].

The references on transplacental transfer of maternal anti-HAV IgG are listed here for the Reviewer’s convenience:

  1. Linder, N.; Karetnyi, Y.; Gidony, Y.; Dagan, R.; Ohel, G.; Levin, E.; Mendelson, E.; Barzilai, A. Decline of Hepatitis A Antibodies during the First 7 Months of Life in Full-Term and Preterm Infants. Infection 1999, 27, 128–131, doi:10.1007/BF02560513.
  2. Derya, A.; Necmi, A.; Emre, A.; Akgün, Y. Decline of Maternal Hepatitis A Antibodies during the First 2 Years of Life in Infants Born in Turkey. The American Journal of Tropical Medicine and Hygiene 2005, 73, 457–459, doi:10.4269/ajtmh.2005.73.457.
  3. De Silvestri, A.; Avanzini, M.; Terulla, V.; Zucca, S.; Polatti, F.; Belloni, C. Decline of Maternal Hepatitis A Virus Antibody Levels in Infants. Acta Paediatrica 2002, 91, 882–884, doi:10.1111/j.1651-2227.2002.tb02849.x.
  4. Lieberman, J.M.; Chang, S.; Partridge, S.; Hollister, J.C.; Kaplan, K.M.; Jensen, E.H.; Kuter, B.; Ward, J.I. Kinetics of Maternal Hepatitis A Antibody Decay in Infants: Implications for Vaccine Use. The Pediatric Infectious Disease Journal 2002, 21, 347.

Reviewer’s Comment 6: The explanation of figure 3 is not clear. Please give more details.

Authors’ Response 6: Further details on Figure 3 have now been provided in lines 242 – 249 which read as follows:

When comparing those 1 – 12 years of age with (HIV, HEU, and HIV-E subgroups) and without (HUU subgroup) perinatal HIV exposure, the proportion of those with natural immunity was highest among the HIV-exposed population. Seropositivity of anti-HAV was 51.6% (16/30) and 29.2% (47/161) in the HIV-exposed compared to the unexposed subgroup, respectively (p = 0.015) (Figure 3). That for anti-HAV IgM was 6.7% (2/30) in the HIV-exposed subgroup with no cases of acute infection detected among HUU children. The highest proportion (70.8%, 114/161) of susceptible children was found in the HUU subgroup (p = 0.015) (Figure 3).”

Reviewer’s Comment 7: The quality of figure 4 is not good, and numbers are not visible well; please provide a higher resolution figure.

Authors’ Response 7: A higher resolution of Figure 4 has been provided in the revised manuscript in lines 261 - 264.

Round 2

Reviewer 2 Report

Comments and Suggestions for Authors

The authors addressed satisfactorely the concerns.

Reviewer 3 Report

Comments and Suggestions for Authors

The changes made to the manuscript improve its quality, and I think it can be accepted.